# Using machine learning pipeline to predict entry into the attack zone in football

**Leandro Stival**[1], **Allan Pinto**[2], **Felipe dos Santos Pinto de Andrade**[3], **Paulo Roberto Pereira Santiago**[3], **Henrik Biermann**[4], **Ricardo da Silva Torres**[5], **Ulisses Dias**[1]*

**1** School of Technology, University of Campinas, Limeira, São Paulo, Brazil, **2** Brazilian Synchrotron Light Laboratory (LNLS), Brazilian Center for Research in Energy and Materials (CNPEM), Campinas, São Paulo, Brazil, **3** School of Physical Education and Sport of Ribeirão Preto, University of São Paulo (USP), Ribeirão Preto, São Paulo, Brazil, **4** Institute of Exercise Training and Sport Informatics, German Sport, University Cologne, Cologne, Germany, **5** Department of ICT and Natural Sciences, NTNU—Norwegian University of Science and Technology, Aalesund, Norway

* ulissesd@unicamp.br

**Data Availability Statement:** Data are available on Figshare (DOI: 10.6084/m9.figshare.19222746). Code are available on GitHub (https://github.com/lstival/soccer_graph_classification).

## Abstract

Sports sciences are increasingly data-intensive nowadays since computational tools can extract information from large amounts of data and derive insights from athlete performances during the competition. This paper addresses a performance prediction problem in soccer, a popular collective sport modality played by two teams competing against each other in the same field. In a soccer game, teams score points by placing the ball into the opponent's goal and the winner is the team with the highest count of goals. Retaining possession of the ball is one key to success, but it is not enough since a team needs to score to achieve victory, which requires an offensive toward the opponent's goal. The focus of this work is to determine if analyzing the first five seconds after the control of the ball is taken by one of the teams provides enough information to determine whether the ball will reach the final quarter of the soccer field, therefore creating a goal-scoring chance. By doing so, we can further investigate which conditions increase strategic leverage. Our approach comprises modeling players' interactions as graph structures and extracting metrics from these structures. These metrics, when combined, form time series that we encode in two-dimensional representations of visual rhythms, allowing feature extraction through deep convolutional networks, coupled with a classifier to predict the outcome (whether the final quarter of the field is reached). The results indicate that offensive play near the adversary penalty area can be predicted by looking at the first five seconds. Finally, the explainability of our models reveals the main metrics along with its contributions for the final inference result, which corroborates other studies found in the literature for soccer match analysis.

## Introduction

Soccer is a collective sports modality of global importance and has attracted the attention of billions of audiences/people worldwide. Roughly speaking, the number of goals scored by each team determines the winner of a soccer match. Thus, a common strategy carried out by teams

**Funding:** Fundação de Amparo à Pesquisa do Estado de São Paulo #2019/17729-0. The funders had no role in study design, data collection and analysis, decision to publish, or preparation of the manuscript.

**Competing interests:** The authors have declared that no competing interests exist.

for scoring goals consists of dominating ball possession and seeking offensive positioning based on the ball's movement in the opponent's defense field.

According to Wright et al. [1], most finishing moves occur within or near the penalty area, which is the region with the highest shot conversion rate. Therefore, reaching this region is an important step toward achieving goal attempts. In addition, Ruiz et al. [2] also showed the importance of this region of the field, showing that teams that enter more into the opponent's area are more likely to win matches.

Since getting close to the opponent's defense field is essential for scoring, some studies were conducted toward understanding strategies for reaching this objective. Marchiori et al. [3] analyzed teams' efficiency a few moments before the scoring, in which the authors demonstrated the importance of controlling certain positions on the field and the team's behavior during finishing moves. Wright et al. [1] concluded that fast plays (with four passes or less) have high chances of converting finishing moves into goals, indicating that the number of passes has an important impact on the final score of the match. The authors also concluded that the number of attackers in the penalty area causes a positive impact on scoring goals. Other influential factors are: the shot's location, the position of the goalkeeper, and the shot type. Another essential aspect that improves team scoring is player formation [4]. Moura et al. [5] observed that teams tend to keep a more compact formation (players closer to each other) when defending.

Although the individual movement of players plays an important role during the matches, such as dribbling and taking shots at the goal, mainly in one-to-one confrontations, this work will focus on the collective behavior of teams, more precisely in attacking and defensive formations. Analysis of teams' behavior has been an object of investigation in several studies. Boboota and Kaur [6] predicted the final score of soccer matches by defining important metrics/variables that lead to a reliable prediction. Frencken et al. [7] use the centroid of teams in the field (the central point considering the position of players from the same team) to identify temporal patterns of teams' movement.

Another approach to visualize and analyze teams' behavior in soccer matches relies on taking advantage of complex network analysis, in which vertices represent players and edges represent relationships between them, usually reflecting the possibility of passing the ball. Rodrigues et al. [8] investigated teams' behavior performing a visual rhythm analysis along with time series of one-match network metrics. Clemente et al. [9] used the centrality of the players to determine which team would win a match, while Malta et al. [10] used the centrality and proximity of the vertices (players) to identify the relationship between these metrics and the chance of scoring a goal. None of those studies investigated the use of complex network metrics in combination with machine learning models with the intent to classify ball possessions.

This research takes advantage of complex network analysis to answer the following research questions: (i) Can we predict if an attacking team will reach the fourth quarter of the field (attacking zone) by considering teams' movements during the first five seconds after a team takes control of the ball?; and (ii) Could complex network analysis provide a set of meaningful features to predict if the ball will reach the fourth quarter of the field before losing control of the ball? We model the problem using a supervised learning approach considering well-established classification algorithms. Our methodology also considers the use of explainable machine learning algorithms to provide insights regarding the importance of each feature to the final predictions produced by the prediction models built in this study.

## Related work

Several works used different approaches to do analysis in the football scenario. We can separate them into three general categories. The first one is the ones that take into consideration

the position of players in the field and their moving patterns. In this approach, Wright et al. [1] proposed a division of the attacking zone into eight parts, and evaluated the rate of scoring for each part. In another work, Vilar et al. [11] analyzed the moving patterns of players in a specific region of the field to predict future values of ball possession. In a similar way, Barreira et al. [12] studied how the behavior of a team might differ depending on the region of the pitch a team got the ball possession. Moura et al. [13], in turn, showed the importance of positioning regarding defense and attack.

The second category of studies for football is the ones that focus on interactions among players. A work in this category is the one by Merlin [14], in which a study of the passes is performed and a Support Vector Machine (SVM) classifier is used to label the passing difficulty. Another work in this category is the one by Frencken and Lemmink [7]. Their study used the position of players to determine a centroid of the team, and with that predict chances of scoring. Another study explored spatiotemporal features associated with player tracking data different from ours, to solve similar problems. For example, Spearman W. [15] used this kind of data to predict the chances of a player creating a scoring situation.

Finally, in the third category, we have studies that use graphs as a way to obtain information from the teams. Rodrigues et al. [8] studied how matches could be represented in graphs and how complex network metrics could be generated from these graphs, also encoding temporal properties. Grund [16] identified a relationship between the importance of a player and the set of passes of a team, which was defined as the centrality of the player. With that, the author noticed that the higher the centrality, the more passes were performed in general. Clemente, Sarmento, and Aquino [9] performed a case study, identifying different types of positions for the player with the highest values of centrality, depending on the championship used for the study.

Besides the discussed categorization, that are also studies that make use of artificial intelligence techniques to make various analyses with football data. One of the most prominent of those is the ones using artificial neural networks and deep learning. In the following, we present some recent approaches with that focus.

The use of deep convolutional networks for soccer analysis has already been introduced by Wagenaar et al. [17]. They exploited convolutional networks; image processing capabilities by transforming position data into 2D images of $256 \times 256$ pixels with different color codes for players, goalkeepers, and goals. From these images, they predict whether a possession ends with a scoring opportunity or a turnover. In their experiments, they compared various convolutional architectures and they found that GoogLeNet yielded 67% accuracy to achieve superior performance.

Deep reinforcement learning in soccer has been applied by Liu et al. [18]. The authors introduced a framework with two separate architectures for the home and away teams to estimate their probabilities of scoring. Therefore, they processed a broad range of descriptive features, such as the position and type of the event or the time remaining in a game. For this task, they proposed to use deep long short-term memory network architectures for both teams separately. Based on the network output, the authors defined a metric to measure the goal impact of individual players. They found that their metric outperforms similar predictors for goals and assists over a season.

The approach by Liu et al. has been extended by Rahimian et al. [19] to optimize behavior in critical situations. The authors similarly described possessions using a broad range of hand-crafted features. Based on this feature description, they learned to predict the behavior of players using a combination of convolutional and long short-term memory neural networks. Moreover, they trained a logistic regression model to predict the scoring probability for a given shot. They then combined the two models to derive an optimal behavior policy that

maximizes expected goals and they use this metric to evaluate real-world actions by comparing them to that policy. The authors found that using the optimal policy, the mean expected goals value of -0.1 of the real-world policy can be increased by 0.55.

Using data from the behavior of teams, such as the number of passes and possession of the ball time, a classifier was trained to predict the chances of winning a match in the Belgian football championship in the work of Geurkink et al. [20]. In that work, the authors showed the importance of shots on target next to the penalty box, a result that reinforces the importance of our work (predict the arrival next to the penalty box). Another point in common with our project is the representation of the importance of features through shapley values.

Video processing with deep learning is usually applied in the tracking of events during the game, for example, in the work of Liu et al. [21], where a 3D bidirectional convolutional network was used to produce a model able to identify the initial point of an event (e.g., shot, cornet kick, foul, and goal), verifying that the pass has a big influence in the current state of a video.

Convolutional networks were also utilized in the work of Zheng [22]. Their study focused on how to predict a player's motion trajectory using a group of 12 cameras. Through them, it was possible to capture the position of the player regardless of where they are on the soccer field, keeping the temporal consistent, given that this approach merged all the captures in a current time with the previous frames of videos.

Data from 118 first-division football matches of the Dutch Eredivisie was collected over four seasons to develop the work proposed by Goes et al [23]. Their work classified ball possessions as successful or not, based on the chances of making a goal. The biggest difference between this work and other works in the literature is the way teams are split up into subgroups. These subgroups distinguish between attackers, midfielders, and defenders in a spacetime form. For each frame from which the information was extracted, the centroid of the position of each of the groups was then calculated.

The work of Tureen and Olthof [24] used 906 matches of the English championship, between the men's and women's championships. Taking the actions of both teams to estimate the probability of a shot being converted into a goal and the participation of each player in the development of the play. Their work used specific variables for shots, such as the preferred foot for the player to kick, the position of the shot, and the position of the goalkeeper. These data were provided to a Generalized Linear Mixed Model (GLMM), and their results were compared with XGboost models, showing that it was a good method to make predictions.

Also using data from the Premier League, Bilek and Ulas [25] carried out their research to identify which performance variables of teams and players can predict the outcome of the match. In their experiments, machine learning techniques, such as K-means and decision trees, were used along with classical statistical techniques, such as ANOVA. The quality of the opposing team was presented as a factor capable of indicating the probable result of the match, given that in matches with teams of balanced level, the one that scores the first goal usually comes out victorious.

Data from La Liga games between 2012 and 2013 composed of 20 teams were used by Brooks et al. [26]. The dataset was used to train a model capable of predicting whether a sequence of passes belonged to a particular team. From the data that was extracted, the number of passes and where on the field they were performed were counted, so that heat maps could be made to show the style of each team. After the pass sequences were created, a K-nearest neighbor (KNN) was trained to classify the training sequences with an accuracy of 87% after 2000 training learnings.

The data on the players' position and the ball captured at a rate of 20 frames per second were used in Dick and Brefeld's work [27]. The goal was to train a model using reinforcement

techniques to identify dangerous situations, which were defined as possession of balls that get inside the penalty area. The training was done by giving each play a score so that the model would give the highest scores to the highest-risk plays. This way, it was learned how teams' tactics relate to each other.

The recent papers cited above have different interests than ours. However, they are still important because confirm how powerful it is the application of convolutional neural networks. The reuse of pre-trained models like DenseNet and EfficientNet increases the speed of training and development of networks in places where originally they are not implemented from scratch.

## Materials and methods

This study aims to answer two important questions from both computer science and sports science perspective. From the computer science perspective, this study introduces a new methodology for predicting the chances of the attacking team reaching the attacking zone, by using both teams' behaviors during the first five seconds after the control of the ball is taken. Generally, only the attacking team has been utilized in analyses, but recent work shows that the defense team is important as well [28]. Thus our methodology opens new research lines for modeling collective behavior over time. From the sports science perspective, the characterization of which features play an important role in predicting chances of scoring may help coaches, technical committees, and researchers in sports science to find new training procedures for enhancing teams' tactical behavior, also our project was approved by the Research Ethics Committee of the State University of Campinas, protocol CAAE 56582616.8.0000.5404. And since all those present are of legal age, the need for consent was waived by the ethics committee, due to the video recording being done with proper cameras installed in the stadium in professional games.

Fig 1 presents the proposed methodology, in which the first step comprises detecting and tracking players toward estimating their *(x,y)*-coordinates on the field. This first step used DVideo software [29] for estimating players' and ball locations on the field. Next, we use complex networks to represent players (vertices) and the chances of passing the ball to teammates (edges) for each team at each instant of time *t*, measured in milliseconds. Then, the concept of

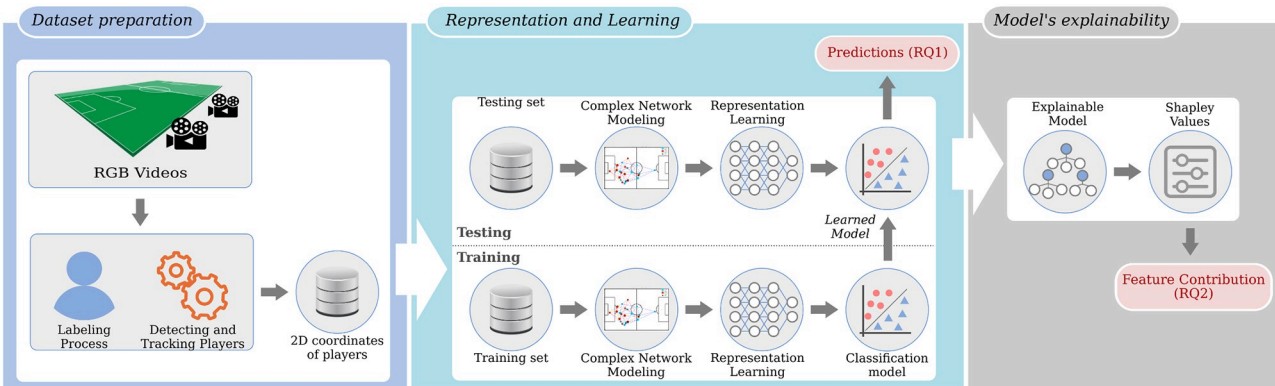

**Fig 1. Proposed method.** Given a set of soccer videos, the proposed method consists of detecting and tracking the players in the field toward obtaining their (x,y)-coordinates. Then, we divided the tracking data into training and testing subsets, using the k-fold cross-validation protocol. Next, we model the interaction between players using complex networks, in which the nodes represent players, while edges the chances of passing the ball to teammates. Thus, we extract representations for the built complex network and use the visual rhythms technique to summarize such features for a time window. Finally, we use the visual rhythms maps to build a classifier for predicting the changes of an attack move reaching the attack zone, and also an explainable method to estimate the contribution of input features.

visual rhythms [8] was employed to summarize different metrics extracted from complex networks in order to characterize graph properties. The visual rhythms extracted from the complex networks are used for classification models to predict the chances of an attacking team reaching the attacking zone, which is related to the first research question raised in this study. Finally, we adopted the use of explainable artificial intelligence (XAI) methods to estimate the contributions of input features and then discover insights about which features contribute most to the final score of our prediction model. In the next sections, we explain in detail the main steps of the proposed method.

## Detection and tracking of players

This study used the dataset freely provided by Merlin *et al*. [14] comprising ten matches from the Brazilian championship. Players' locations were estimated using the software DVideo [29], which takes as input videos generated by calibrated cameras and allows setting the (x,y)-coordinates of players on the field at a frequency rate of 30Hz. The main events (like passes, faults, penalty kicks, and others) were annotated by specialists in sports science as reported by Merlin *et al*. [14].

We arranged the dataset considering maximal intervals of time in which a team holds control of the ball, named "ball possession intervals (BPI)." In other words, BPI comprises the entire period of time in which a team acquires possession of the ball until it loses control of the ball to the other team. We further split BPI into three parts, as illustrated by Fig 2: (i) the Feature Extraction Time-Window (FETW) represents the first 5 seconds of the BPI, which is used to extract the features served as input to the models; (ii) the Lag is the region in the middle of a BPI and it has a flexible time size, which depends on the BPI size in seconds; and (iii) the Target comprises the last 5 seconds of BPI. The Target is the period of time we use to assess whether or not a team reached the fourth quarter of the field (attacking zone) to label the BPI as success (attacking team reaches the attacking zone) or failure (attacking team could not reach the attacking zone). Other sizes of FETW were tested (3 and 7 seconds), but we choose 5, which proved to lead to superior effectiveness.

## Complex network modeling and feature representation

We have previously mentioned that the dataset stores the (x,y)-coordinates of all players on the field at a frequency rate of 30 records per second. For each record, we generate a graph $G = (V, E)$, where $V$ represents the set of players, and $E$ represents the possibility of a pass between the players. We say that the possibility of a pass between two players $a$ and $b$ exists if two conditions are satisfied: (i) there is no player from the same team between $a$ and $b$; and (ii) there is no player from the opponent team 50 centimeters or fewer close to $a$ or $b$, the connections between $a$ and $b$ were determined using *Delaunay* triangulation.

We extract features from the graphs to fully characterize team behaviors. Fig 3 depicts a graph created from the (x, y)-coordinates of each player. Note that players are vertices in the graph, and the edges represent whether a player can pass the ball to a teammate or not. Observe that each vertex is surrounded by a circle. The circle sizes proportionally indicate the metric called Betweenness Centrally of each player, which allows one to observe the metric distribution in the pitch.

**Metrics.** The following metrics were extracted from graphs. These metrics were chosen based on other studies that analyzed soccer matches using a complex network perspective [8].

• Let $\sigma(s, t)$ be the number of shortest paths from $s$ to $t$, and $\sigma(s, t|v)$ be the number of shortest paths from $s$ to $t$ passing through $v$ [30]. The **Betweenness centrality** of a vertex $v$ is the sum

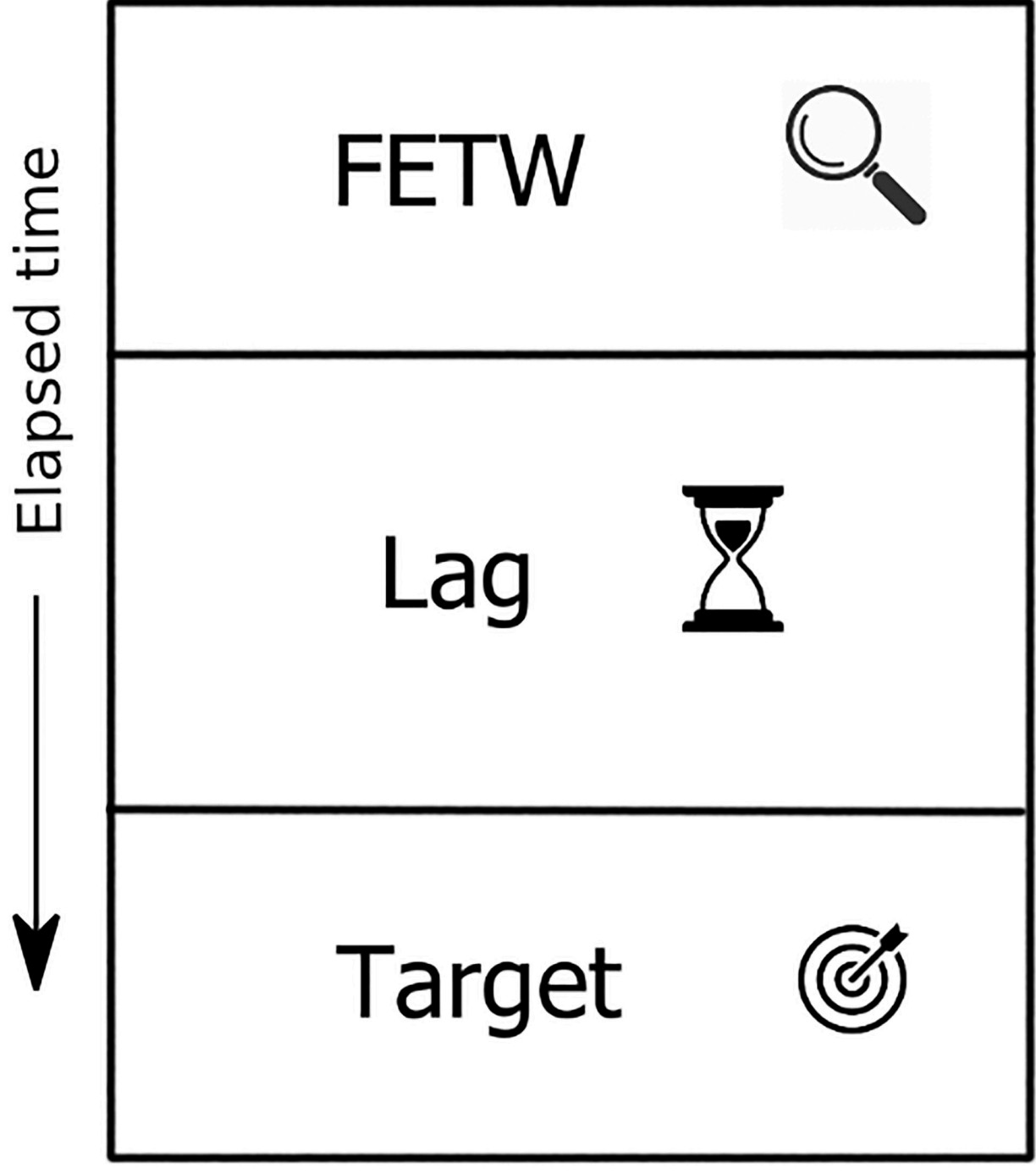

**Fig 2. Division of BPI possession windows.**

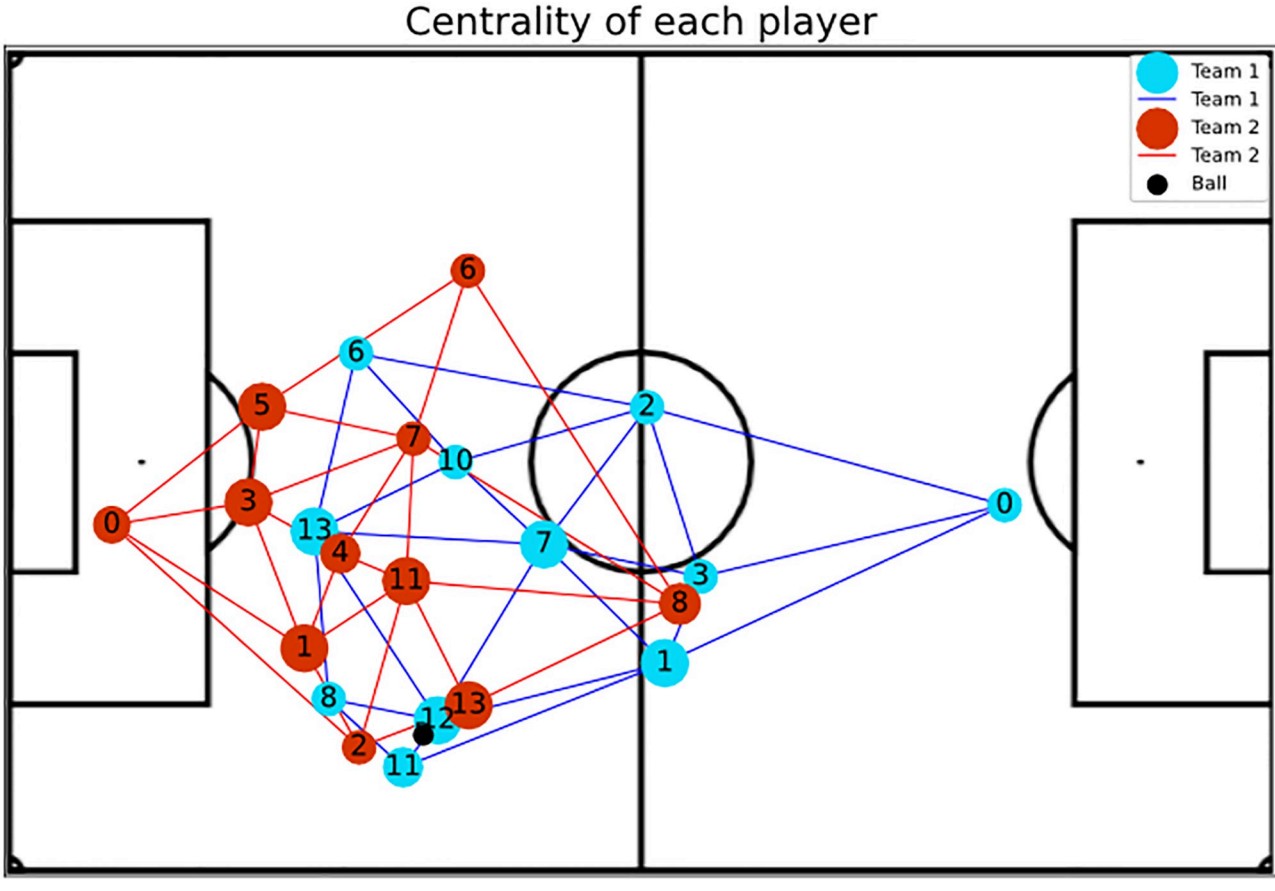

**Fig 3. The figure depicts a graph built from the (x, y)-coordinates of each player.** Each vertex has a circle surrounding it, and the circle sizes aid to compare the betweenness entropy of a vertex with the others, where the larger the circle, the greater the betweenness entropy.

of $\sigma(s, t|v)$ over all distinct pairs of vertices $s$, $t$ divided by the total number of shortest paths.

$$c_B(v) = \sum_{s,t \in V} \frac{\sigma(s, t|v)}{\sigma(s, t)}, \text{ where : } \begin{cases} \sigma(s, t) = 1, & \text{If } s = t, \\ \sigma(s, t|v) = 0, & \text{If } v \in s, t \end{cases} \quad (1)$$

A common normalization of the betweenness centrality is shown in Eq 2, where $n$ is the number of vertices.

$$c_{Bn}(v) = \frac{c_B(v)}{(n-1)(n-2)} \quad (2)$$

- The **Eccentricity** of a vertex $v$ is the maximum distance between $v$ and some other node. The distance $d(u, v)$ between two vertices $u, v$ is defined as the length of the shortest path between $u$ and $v$.

$$e(v) = max(d(v, v_j)), \forall v_j \in V \quad (3)$$

- The **Global Efficiency** of a vertex $v$ measures the information transmission capacity of the graph through $v$ and indicates how efficient it is to send information among vertices. The efficiency is proportional to the inverse of the distance between $v$ and other vertices $j$ [31].

Eq 4 shows the formula, where $n$ is the number of vertices.

$$E(v) = \frac{\sum_{j \neq v}^{j \in G} \frac{1}{d(v,j)}}{(n-1)} \tag{4}$$

- The **Local Efficiency** of a vertex $v$ is computed for a subgraph $\hat{G}_v$ that contains the neighbors of $v$ and their edges [31]. The local efficiency measures fault tolerance by assessing the impact on local communication (direct neighbors) when a vertex and all its edges are removed. Eq 5 offers a formal definition, where $deg(v)$ is the degree of $v$.

$$E_{loc}(v) = \frac{1}{deg(v)} \sum_{v_j \in \hat{G}_v} E(v_j) \tag{5}$$

- The **Vulnerability** of a vertex $v$ uses the global efficiency and the local efficiency of $v$ to compute the drop in efficiency after the removal of $v$:

$$V(v) = 1 - \frac{E_{loc}(v)}{E(v)} \tag{6}$$

- The **Clustering Coefficient** characterizes the presence of triangles (loops of order three) formed with a vertex $v$. Let $T(v)$ be the number of triangles formed with $v$ [32], the clustering coefficient is computed as shown by Eq 7.

$$C(v) = \frac{2T(v)}{deg(v)(deg(v) - 1)} \tag{7}$$

- The **Entropy** of a vertex $v$ is computed considering the probability $P_h(v, j)$ of reaching a vertex $j$ starting from a vertex $v$ after performing $h$ steps in a random walk [33]. This measure is obtained as shown by Eq 8.

$$E_h(v) = -\sum_{vj} P_h(v,j) \log(P_h(v,j)) \tag{8}$$

- The **PageRank** measures the importance of a vertex $v$ using as source of information the importance of its neighbors [34]. Let $v$ be a node in the graph $G$ and $nei(v)$ be the subset of all neighbors of $v$. The PageRank of $v$ can be obtained by Eq 9, where $q$ is the amortization factor usually set as 0.85 and $n$ is the number of vertices.

$$p(i) = \frac{1-q}{n} + q \sum_{j \in nei(v)} \frac{p(j)}{deg(j)} \tag{9}$$

Note by Eq 9 that the PageRank of a vertex is defined recursively and it depends on the number and PageRank metric of all neighbors. To solve the recurrence relation, PageRank can be computed either iteratively until a convergence condition is met or algebraically. Finally, keep in mind that our definition of PageRank works for undirected graphs, and it slightly changes in the case of directed graphs.

**Metrics from the soccer perspective.** The metrics can be interpreted as follows from the perspective of a soccer match.

- **Betweenness Centrality**: Shows player's impact in play constructions, where higher values indicate a greater availability to participate in further plays and do passes.

- **Eccentricity**: Represents how easy it is for the ball to reach a player considering the entire network. An example was presented in Fig 4.

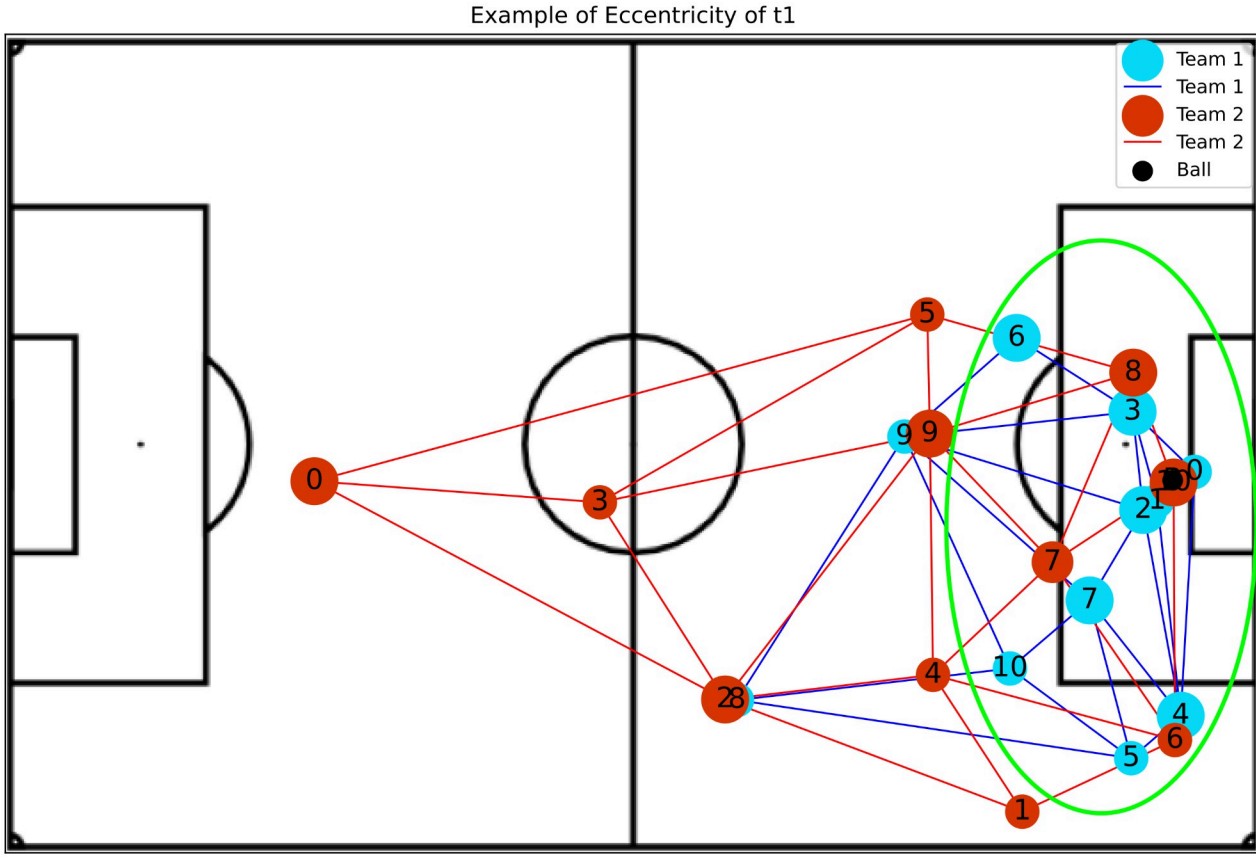

**Fig 4. The image shows a situation with a high value of eccentricity of t1 (team with the ball possession).** Here, it is possible to observe the players are separated in the highlighted green zone, but with many options to make passes.

- **Global Efficiency**: Indicates whether or not players are well positioned so that they become available to make and receive passes.

- **Local Efficiency**: Like global efficiency, it shows the importance of the players in plays, but regarding the impact on nearby teammates. Fig 5 shows an example.

- **Vulnerability**: Shows the impact on the team when a given player is removed, such that negative values indicate a decay in efficiency without the player.

- **Clustering Coefficient**: Indicates the probability of the player making pass triangulation with teammates. This triangulation is very important because allows the team to move with greater speed.

- **Entropy**: Represents the probability of a player receiving a pass from closely related teammates. The higher the value of this metric, the more the chances that this player could participate in the play. Examples of attacking and defense teams are illustrated in Figs 6 and 7, respectively.

- **PageRank**: Shows the prestige of a player to build a play, in other words, the higher this metric, the bigger the participation of this player in passes with other players who also have high participation in plays.

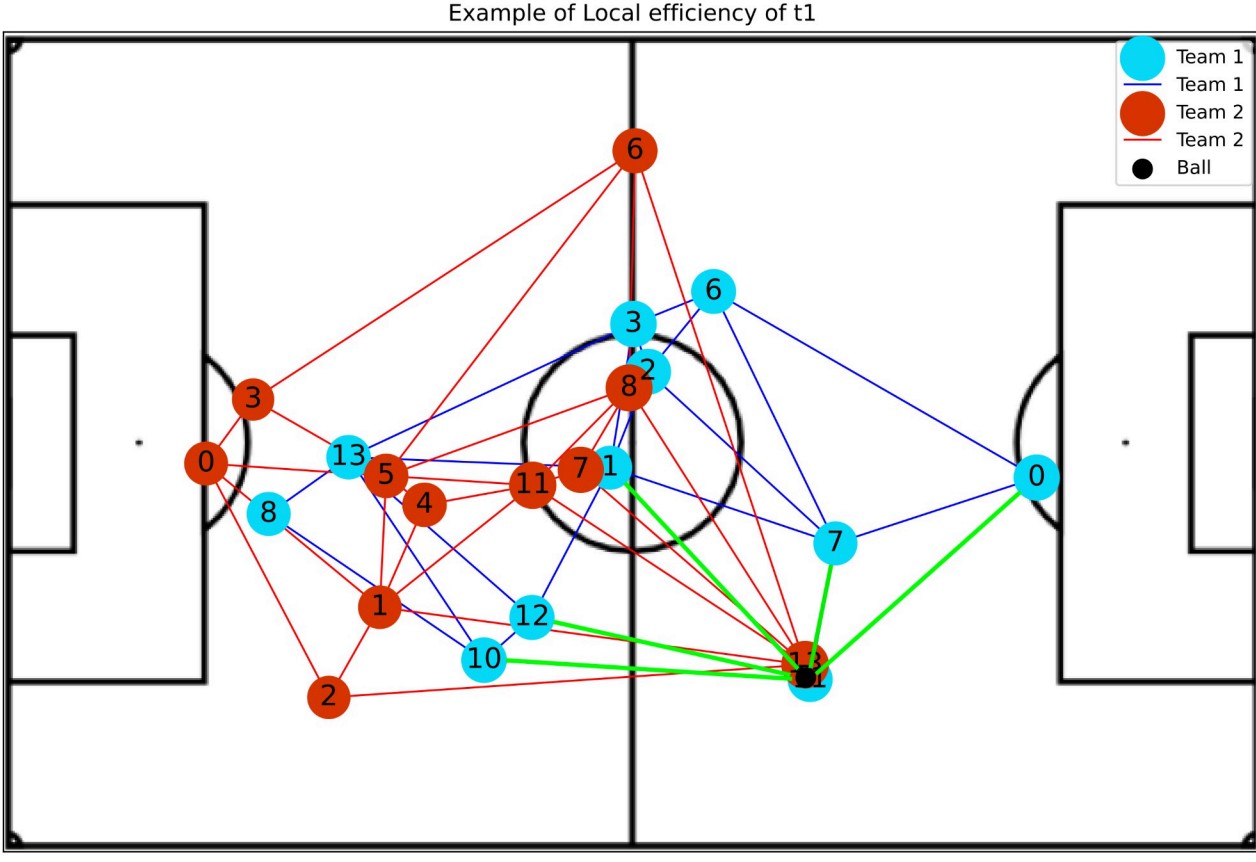

**Fig 5. The image shows a situation with high local efficiency of player 11 at the moment that he received the ball, having five options to make the pass (green edges).** This situation demonstrates that if he was undermarked (unable to receive the ball) this play would not be possible.

**Graph visual rhythm.** The objective of this step is to summarize the information we obtained so far that is encoded in the graph metrics. Notice that we have 15 graphs per second since that was the frequency rate in the input video. Therefore, our data structure is a temporal graph $\mathbb{G}$ defined as a sequence $\mathbb{G} =< G_1, G_2, \ldots, G_T >$, such that $T$ is the number of frames in the FETW portion of a BPI used for training or testing. For each graph in the sequence, we compute a set of graph metrics as previously defined. The metrics are defined for each vertex in the graph, for example, we have a Betweenness Centrality associated with each player (vertex). In addition, to make it easier to later interpret the results, we place the attacking and the defending teams in different maps. That said, for a given metric and a given team we can convert the temporal graph $\mathbb{G}$ into a map $\mathbb{F}$ such that:

$$
\mathbb{F} = \begin{aligned}
&<< F_{0,1}, F_{0,2}, F_{0,3}, \ldots, F_{0,T} >\\
&< F_{1,1}, F_{1,2}, F_{1,3}, \ldots, F_{1,T} >\\
&< F_{2,1}, F_{2,2}, F_{2,3}, \ldots, F_{2,T} >\\
&\ldots\\
&< F_{m,1}, F_{m,2}, F_{m,3}, \ldots, F_{m,T} >>
\end{aligned}
$$

For a soccer match, the value for $m$ is 10 since teams can have up to 11 players. The final step creates an image from $\mathbb{F}$ by normalizing the values and transforming each $F_{i,j}$ into a pixel

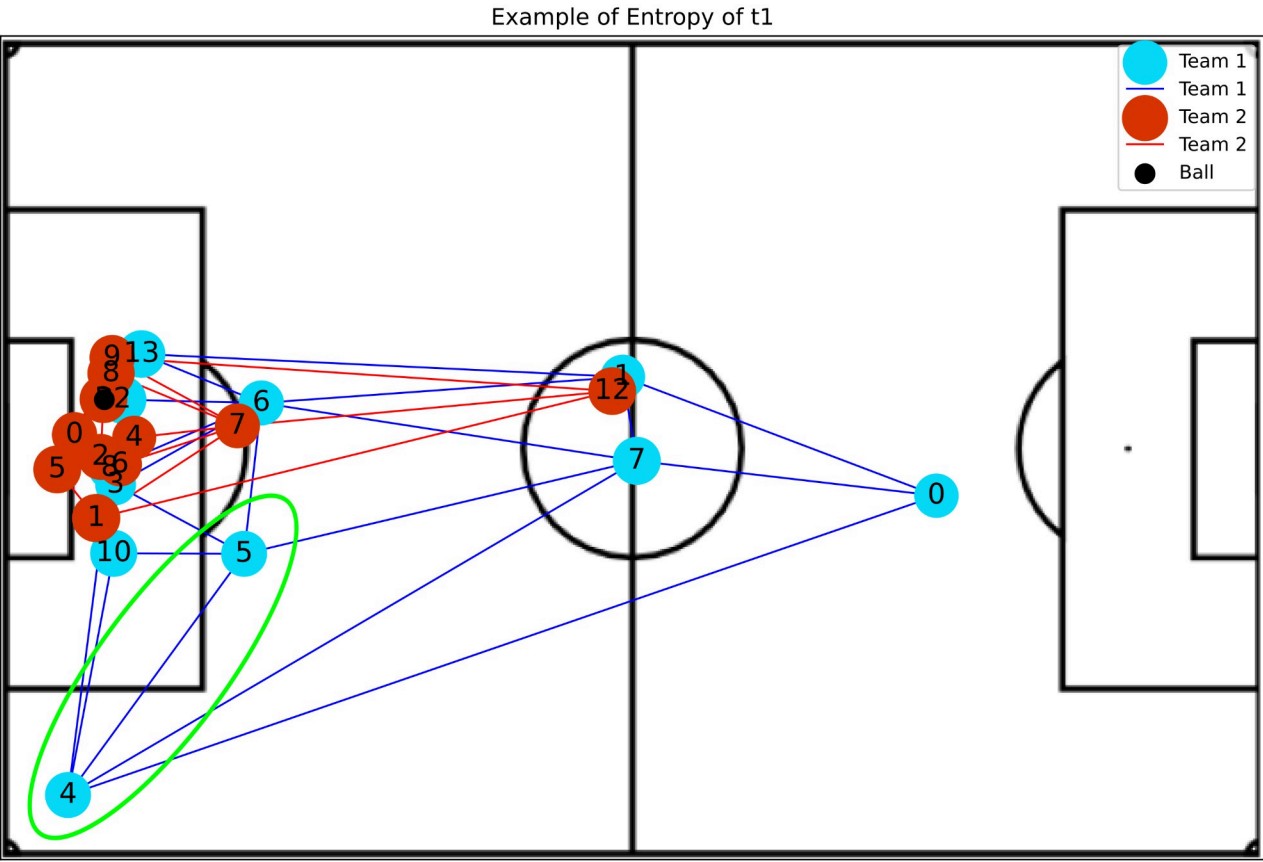

**Fig 6. The image illustrates a high entropy situation for team t1 (with ball possession), where it is possible to observe that players 5 and 4 (the region highlighted in green in the figure) are free of marking and with several options of passes (edges incident on the vertex), thus having a greater chance of receiving passes.**

in the position $(i, j)$. Therefore, the $X$ axis in the graph visual rhythm represents the time (since it came from the frames of the video) and the $Y$ axis represents the players. Fig 8 illustrates how the metrics can be represented in terms of visual rhythms. Fig 8(a) shows the entropy of players 6 and 10 changing over time within a given FETW, and Fig 8(b) encodes the same information in a graph visual rhythm along the $Y$ axis at positions 6 and 10. In a real match, others positions in the $Y$ axis would be filled with information from other teammates. With the metrics computed for all players, the complete visual rhythm will look like the one depicted in Fig 9, which contains 11 players on the $Y$ axis and 5 seconds on the $X$ axis.

This process was repeated for both teams and for each metric, which generated several visual rhythms like those presented in Fig 11. They were joined together in a single image where each metric is a channel in the image by gathering visual rhythms for the eight metrics and arranging them according to Fig 10, following the order of channels: (1) Centrality; (2) Clustering coefficient; (3) Eccentricity; (4) Entropy; (5) Global efficiency; (6) Local efficiency; (7) PageRank; and (8) Vulnerability.

## Machine learning pipeline

For each BPI we have an image that arranges the visual rhythms obtained from the FETW portion, and we also have a label obtained from the Target portion that says whether the BPI

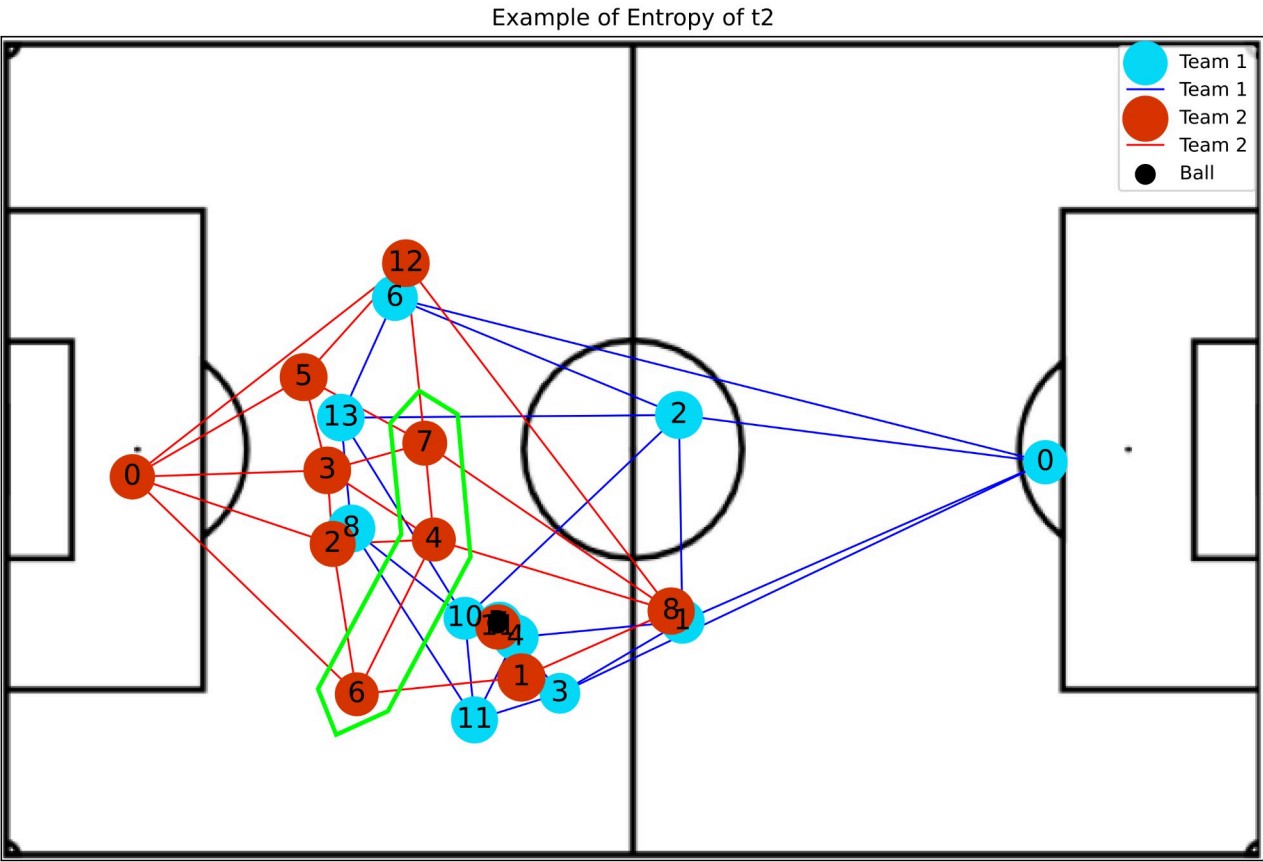

**Fig 7. The image illustrates a high entropy situation for team t2 (without ball possession), where it is possible to observe that players 7, 4, and 5 (the region highlighted in green in the figure) are far from players of the team with possession, thus not marking any opponent, thus facilitating the performance of passes between opposing players.**

corresponds to a success (attacking zone is reached) or failure (attacking zone is not reached). Our next step is to create a machine-learning pipeline using well-established tools.

**Feature extraction.**   We extracted features from visual rhythm images using the deep convolutional neural network called EfficientNetB0 [35] with 6.7 million trainable parameters, which were pre-trained with the ImageNet database. We use the well-known transfer learning mechanism with the fine-tuning process, in which the last layer of the neural network is removed, and the result obtained corresponds to an array of 1.280 floating point values produced by the previous layers. The array is provided as input to a Dense Neural Network (DNN) with 4 fully connected layers with ReLU activation and a decision layer with softmax. During the training, the loss was estimated using balanced accuracy and the Adam optimizer.

**Computational resources.**   The complete training was performed in a computer having Windows 10 (CPU FX-8350-e (8) @ 3.20 GHz; 16 GB RAM; AMD) with GPU (GeForce GTX 1050 Ti; 4096 MB GDDR5; NVIDIA) and took one day (24 hours). Also, to obtain the graph metrics, we used the methods provided by the python library called *networkx*, and it required approximately 4.2 seconds to generate the metrics for each BPI.

**Hyperparameter optimization—Grid search.**   The grid search process allows testing several hyperparameters for the model, so it is possible to select those with the best results. The optimization requires us to split the data in train, validation, and testing, and it is usually

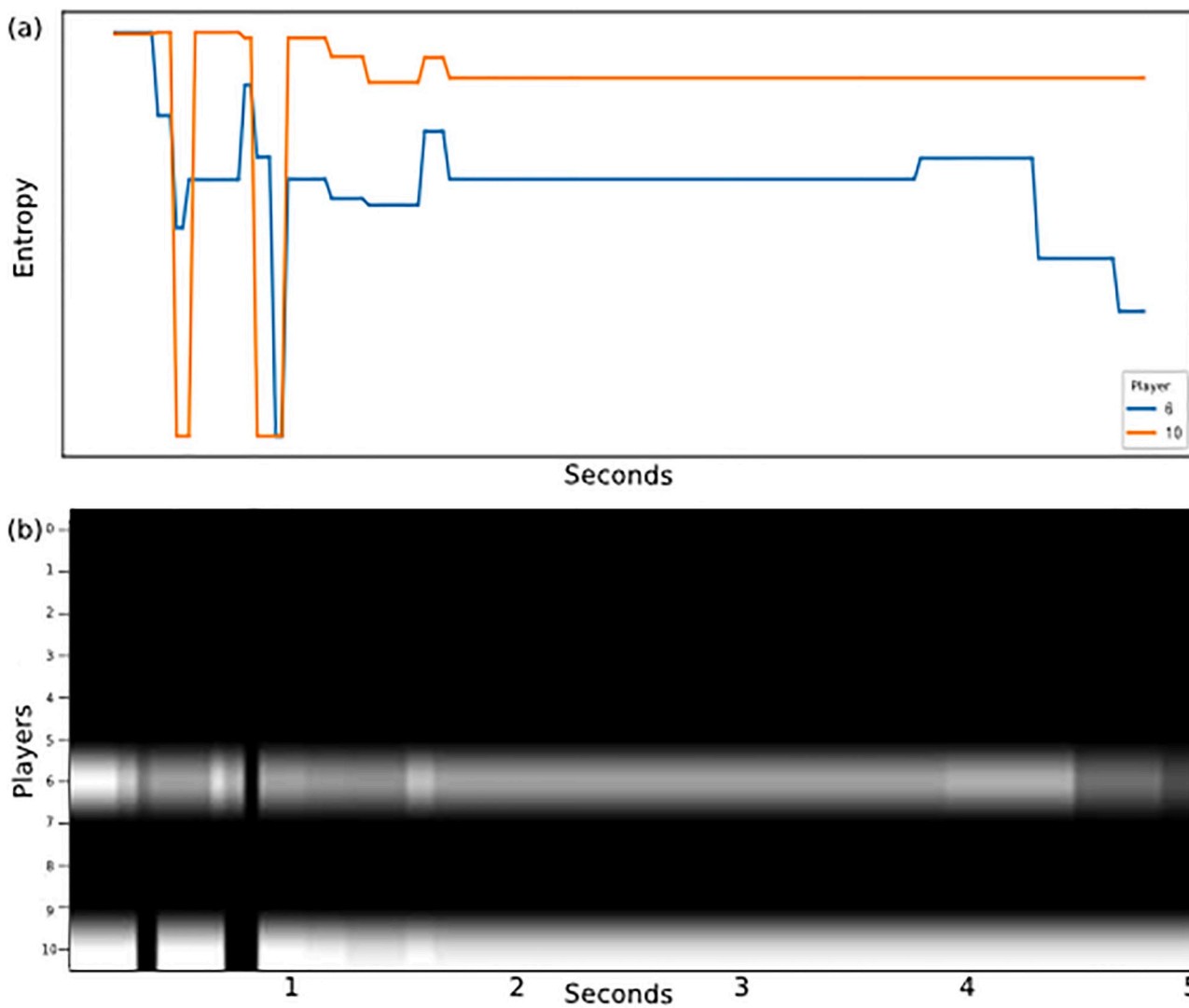

**Fig 8. The figure demonstrate how the entropy values for two players vary over time, where the *X* axis represents the seconds of ball possession and the *Y* axis is the entropy measure at the moment.** (b) shows the same values of item (a), but in the visual rhythm format, where light tones represent higher values of entropy and dark shades the lower values.

recommended to use a cross-validation approach, which we did as follows. A standard soccer match consists of two halves of 45 minutes each, and since we have 10 matches in our dataset, we end up with 20 groups of BPI. We first selected 6 groups as a test dataset and the remaining 14 groups were used for training and validation. We created a set containing all BPIs from the 14 groups and performed cross-validation to find the weights for the network.

**Statistical test.** We executed the machine learning pipeline 10 times to obtain enough data to perform a statistical test. Each run had a different split in the train, validation, and test sets, and we report in the paper the balanced accuracy averaged over the 10 runs presenting a significant difference between rounds, the model results were compared with a random classifier with 50% chance to choose each class, and verified with the no parametric Wilcoxon signed-rank test.

**Metrics contribution.** To analyze the contribution of each metric in the neural network, we used the Python library SHAP (Version 3.8). Our pipeline uses fine-tuning in the

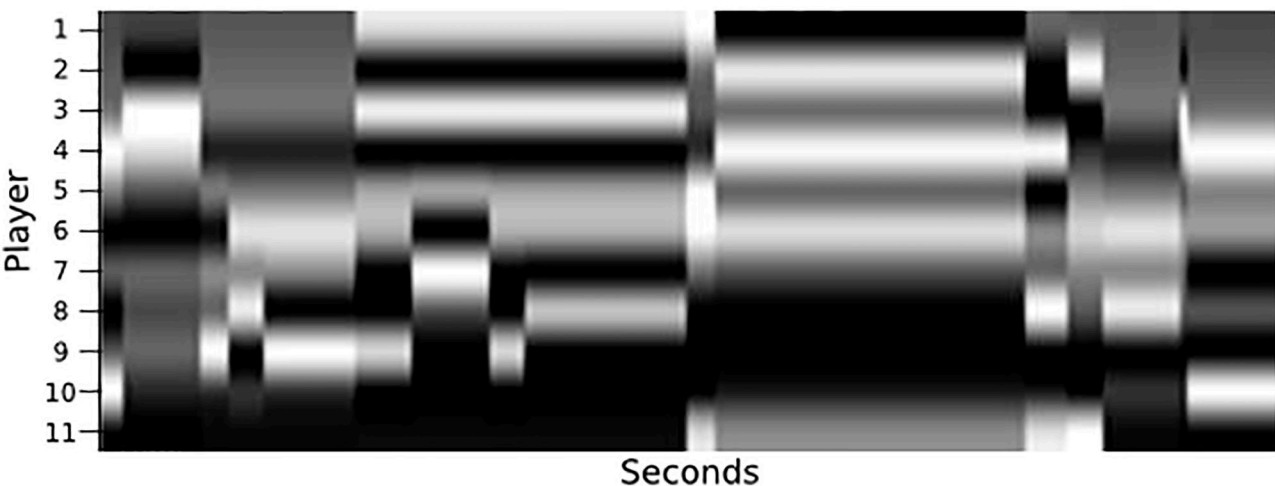

**Fig 9. The *X* axis represents the pass of time and the *Y* axis the players (value of graph metric), the pixels with light colors represent higher values for the player at that moment, while dark tones mean lower values.**

EfficientNetB0. In this way, the input to classification is not the original image, because they have 16 dimensions and the network needs only 3. This transformation is performed on two convolutional layers followed by a resize layer (putting the input in the format that Efficient-NetB0 was trained, having at least $32 \times 32 \times 3$ dimensions).

To solve this problem, the SHAP library has different kernels for explanation purposes, each one is recommended for a different model/classification architecture, for example, to analyze deep neural networks the recommended kernel is *DeepExplainer*, which linearizes the components to identify the model behavior; another kernel commonly used is the *GradientExplainer* to evaluate the impact of pixels in the classification of images by creating a sensitive map that aid identifying the impact of the absence of the pixel. In this work, we used the *Explainer* kernel, which allows for explaining any kind of model by implementing a local linear regression to simulate the result of the original classifier.

Since the kernel *Explainer* can only analyze and explain a single sample at a time, we applied a mask over the result of the SHAP values to generalize the values to all BPIs that reached the fourth quarter of the field, and we performed an absolute sum for each metric. The result of this process has the total percentage of the SHAP values, allowing us to visualize the total impact of each metric.

## Results

### Can we predict if an attacking team will reach the fourth quarter of the field (attacking zone) by considering teams' movements during the first five seconds after a team takes control of the ball?

To answer this question, we exploit complex network metrics and transfer learning with fine-tuning in a Deep Neural Network (DNN) toward building an accurate and reliable prediction model that leads us to predict the chances of an attacking team reaching the attacking zone.

**Visual rhythms.** Fig 10 illustrates the visual rhythm built considering a range of five seconds of an offensive play. To have a holistic representation of teams' behavior, we gather in a single representation eight complex network metrics capable of summarizing different aspects of complex networks built using both the attacking and the defending teams' formations. Fig

(A)

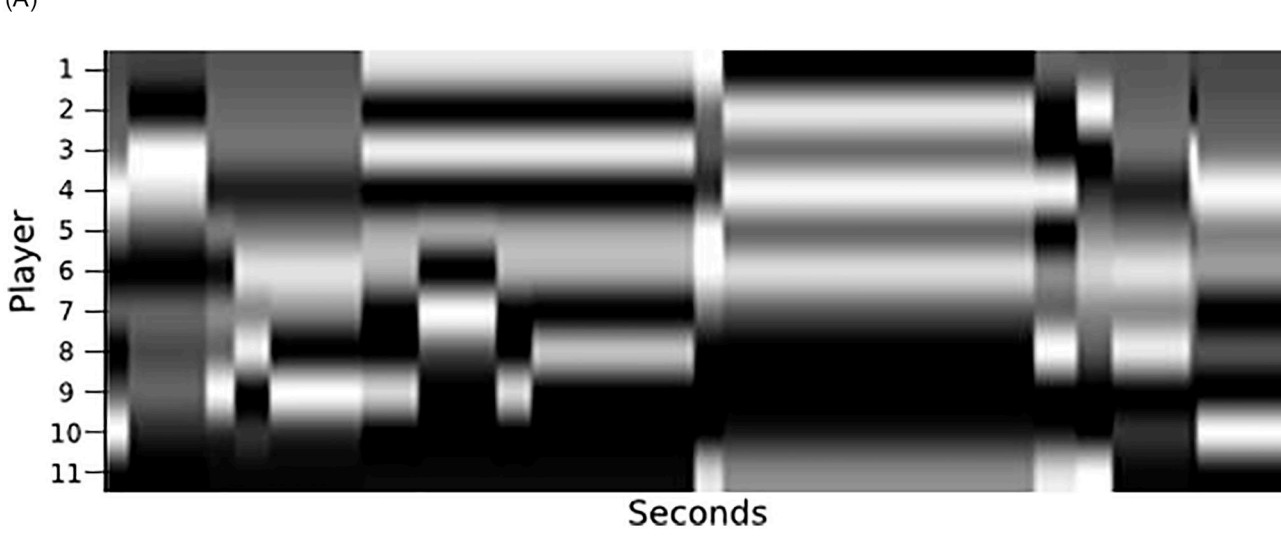

(B)

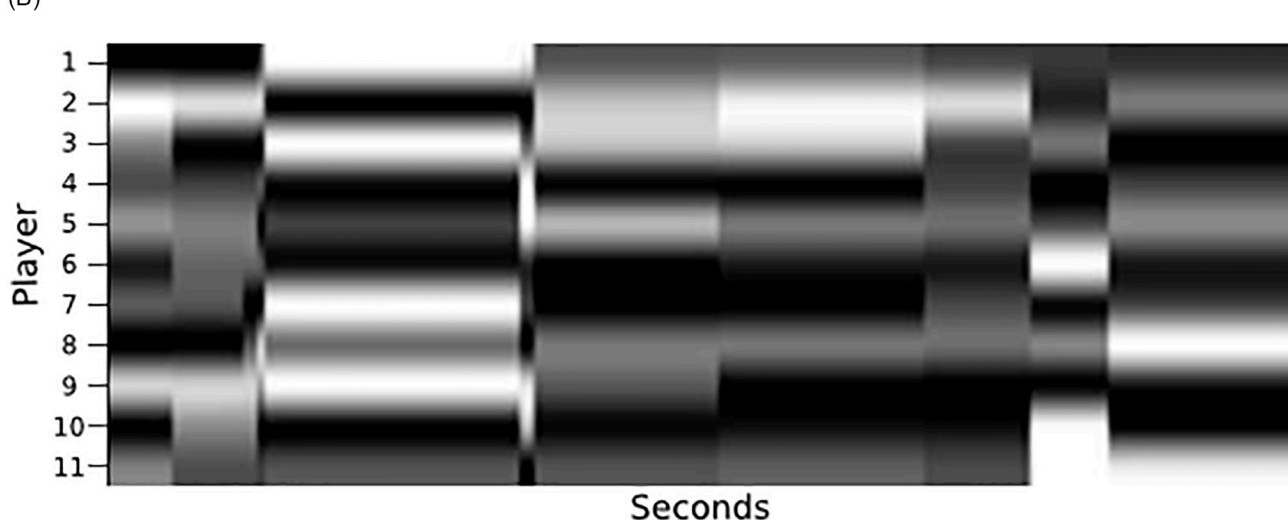

**Fig 10. Visual rhythm built by measuring the entropy of complex networks that represent the attacking (a) and defending (b) teams considering an interval of five seconds.** The x-axis represents the time elapsed, while the y-axis shows the entropy for each node (player) of complex networks. (a) Attacking team (with the control of the ball), (b) Defending team (without the control of the ball).

11 shows how the eight visual rhythms are arranged together to serve as input for classification models.

**Deep representations.** After building visual rhythms based on eight complex network metrics, we extract deep representations taking advantage of deep networks toward producing meaningful representations for further classification. This study adopted the use of *EfficientNet B0* [35] network as a feature extractor, whose weights were set with the ImageNet dataset [36] and fine-tuned with your data. Other deep networks could be used in this process, we choose to use this architecture because it has the best results for transfer learning and uses fewer parameters for that. As a result, we ended up with a feature vector of 1, 280 dimensions for each visual rhythm arrangement produced in the previous step.

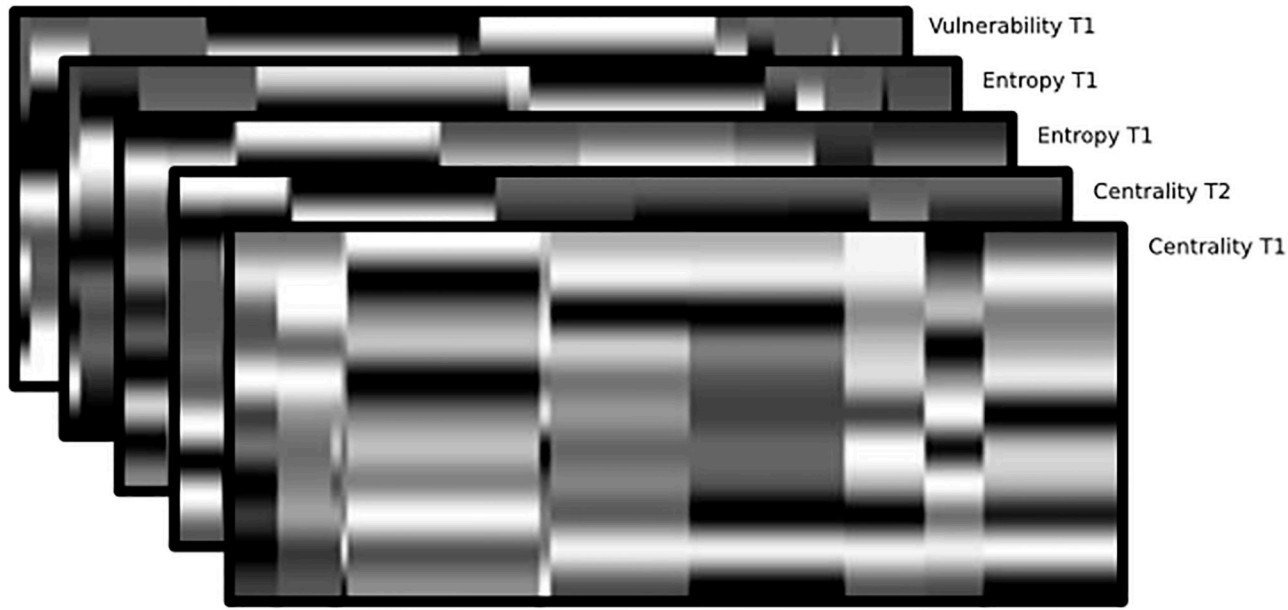

**Fig 11. Visual rhythm arrangement used as input for classification models.** This representation gathers eight different complex network metrics obtained from the attacking and defending teams: (1) Centrality; (2) Clustering coefficient; (3) Eccentricity; (4) Entropy; (5) Global efficiency; (6) Local efficiency; (7) PageRank; and (8) Vulnerability. Where each metric was represented by a channel in the image, in this way each image has sixteen dimensions (one for each of the eight metrics of both teams). So the image is represented with 11 pixels height (one for each player), 167 pixels width (representing time), and 16 dimensions (one for metric).

**Classification models.** To build robust and reliable classifiers to predict whether an attacking team with control of the ball will reach the attacking zone, we investigated the use of the output of EfficientB0 as an input, and we build a Deep Neural Network (DNN) to make the classifications based on the features extracted from the visual rhythm images. This approach generated an architecture with obtained a mean balanced accuracy of 74% and the full results as shown in Table 1.

To complement the results, we calculated the confusion matrices for each round of test data. The results can be observed in Table 2, demonstrating the behavior of the model to classify both classes.

We compare the results of our network in the test dataset with a baseline classifier where classifications were realized with a 50% chance to choose each class. The results for each round were presented in Table 3 with a mean balanced accuracy of 52%. Achieved results demonstrate that our method produces better results than a random classifier.

Having computed these balanced accuracies for the artificial neural network and for the random classifier, we tested the null hypothesis *H0* that verify if there exists a significant difference between the accuracy scores obtained across the rounds between the baseline model

**Table 1. Balanced accuracy of the classifier for each test round.**

| Balanced Accuracy (%) for each Round | | | | | | | | | |
|---|---|---|---|---|---|---|---|---|---|
| **Rounds** | **1** | **2** | **3** | **4** | **5** | **6** | **7** | **8** | **9** | **10** |
| Accuracy | 74% | 79% | 52% | 75% | 77% | 82% | 67% | 75% | 89% | 74% |

**Table 2. Confusion matrices for each test round, where the first row shows the true positive and false negatives values respectively.** And the second row shows the false negative and true negative respectively.

| Confusion matrices for each test round | | | | | | | | | |
|---|---|---|---|---|---|---|---|---|---|
| Round 1 | | Round 2 | | Round 3 | | Round 4 | | Round 5 | |
| 15 | 12 | 18 | 9 | 1 | 25 | 12 | 11 | 13 | 9 |
| 19 | 285 | 25 | 342 | 0 | 292 | 6 | 288 | 11 | 284 |
| Round 6 | | Round 7 | | Round 8 | | Round 9 | | Round 10 | |
| 19 | 7 | 11 | 19 | 13 | 7 | 21 | 5 | 8 | 7 |
| 23 | 275 | 3 | 301 | 37 | 246 | 5 | 278 | 8 | 177 |

(random) and our network, this hypothesis shows us if the results of the network and random classifier coming from the same distribution, which has been rejected when $p < 0.01$. This value is computed with the non-parametric Wilcoxon test, which shows that the differences in the round's results do not occur by chance. That result indicates the variance of the result of the network depends on the matches' data, but generally presents good results, once they are not random.

## Feature interpretation

To understand how complex network metrics impact the prediction, we plot in Fig 12 the influence of each metric considering the team with the ball (referred to as $t_1$), and in Fig 13 we plot the influence of each metric considering the team without the ball (referred to as $t_2$). The influence of the metric is represented by the total SHAP value of each metric in the classifier after being analyzed through the SHAP library, where the values indicate how much the metric helps the classifier to predict the correct class. Considering the defending team, we observe that eccentricity, local efficiency and entropy have the highest percentage. The attacking team also has the same results, but with less percentage and changing the first metric.

The interpretation of metrics can be realized in this way, where eccentricity indicates how easy it is for the ball to reach a player in the network, the choice of this metric for both teams has different implications, which will be explained in the discussion section.

Local Efficiency has a great influence on both teams. This metric indicates how the absence of a vertex would impact the efficiency of your neighbors in the network. High values for this metric indicate that when a player is removed (for example, when they are marked) his neighbors lose an important way to receive the ball, as illustrated in Fig 5.

Entropy in this work represents the probability of a player receiving a pass from a close teammate. High values tell us that player probability will stay free to participate in the play, therefore allowing for a greater variety of plays, as shown in Fig 6.

## Could complex network analysis provide a set of meaningful features to predict if the ball will reach the fourth quarter of the field before the attacking team loses control of the ball?

The ability to explain and interpret prediction model outputs is paramount since the understanding of how models produced a given prediction for a given set of input features could

**Table 3. Balanced accuracy of the classifier for each test round with random classifier.**

| Balanced accuracy (%) for each Round | | | | | | | | | | |
|---|---|---|---|---|---|---|---|---|---|---|
| Round | 1 | 2 | 3 | 4 | 5 | 6 | 7 | 8 | 9 | 10 |
| Accuracy | 50% | 63% | 58% | 51% | 50% | 40% | 51% | 47% | 44% | 63% |

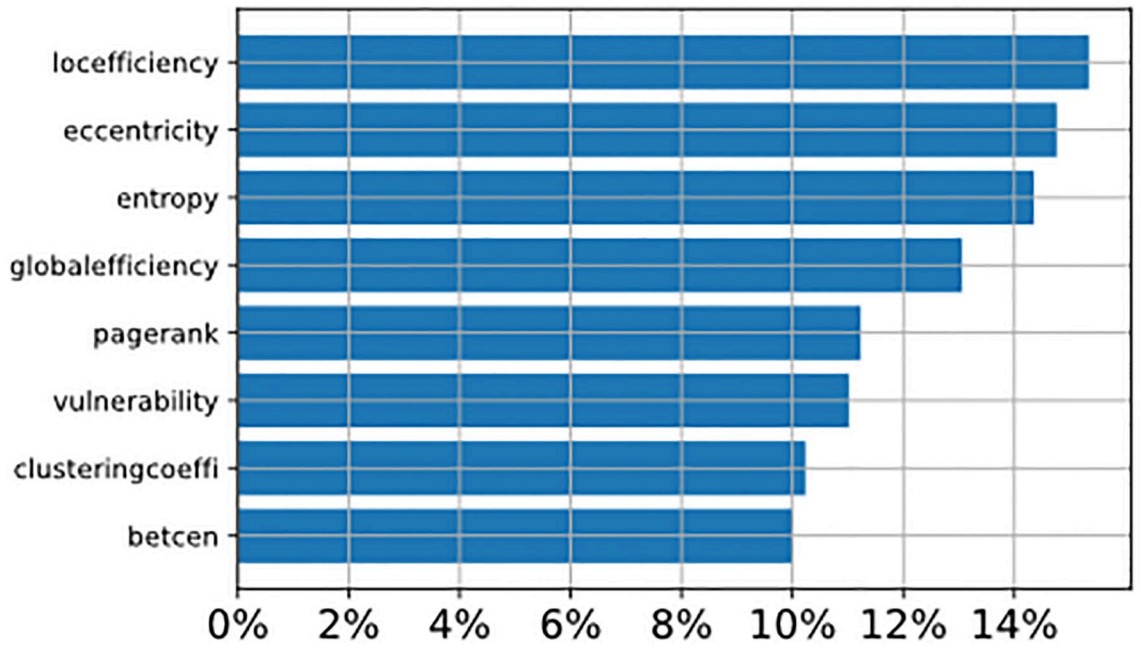

**Fig 12. The figure shows the SHAP values of metrics of the attacking team.**

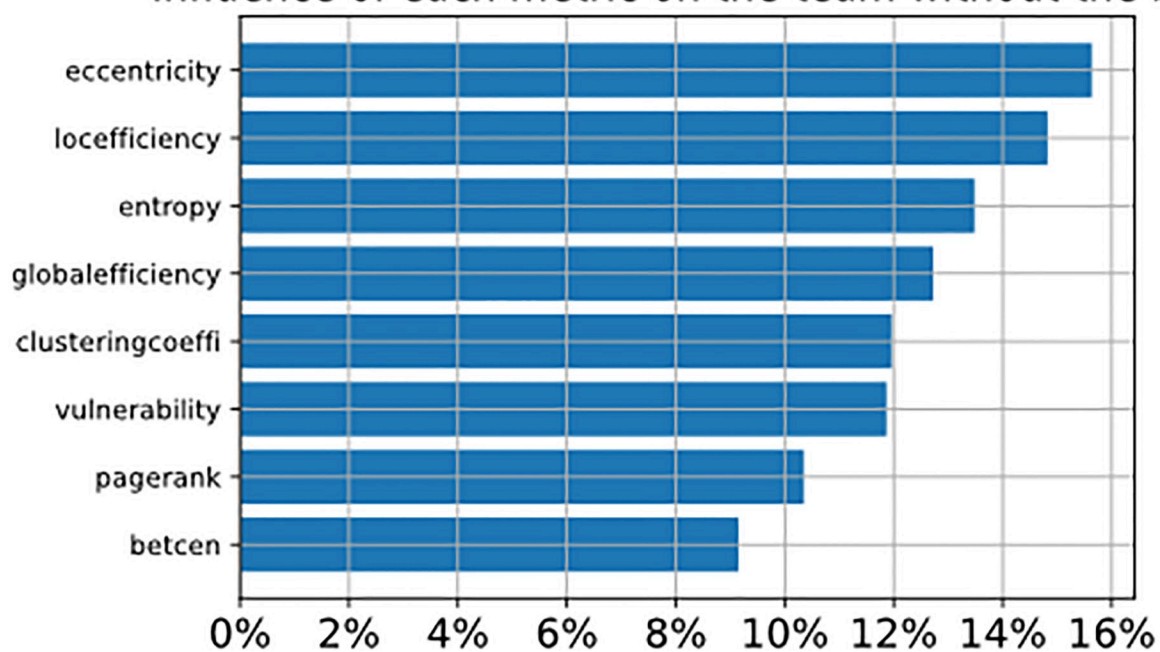

**Fig 13. The figure shows the SHAP values of metrics of the defending team.**

give insights into which features and which range of values lead to a better chance of reaching the attacking zone. Such information could be exploited by coaches, technical committees, and researchers in sports science to improve teams' tactical and technical performance. Currently, the methods devised for explaining machine learning models can be categorized as global and local approaches. Global approaches provide explanations that reveal mechanisms of how models produced outputs' values for the entire datasets. In turn, local approaches provide individual explanations for each data point, which helps humans understand why a given model produced the output for a specific instance [37, 38].

In this study, we used the SHapley Additive exPlanations (SHAP) method [39] to explain models produced to answer the first question. In such a manner, we can estimate the contribution of each input feature for the prediction model's outputs, and thus discover which features are in fact meaningful to predict if the ball will reach the final quarter of the opponent field. In summary, SHAP method estimates feature contributions by computing a shapley value for each input feature, which indicates how much the feature adds to the output value. The concept of shapley value was originally introduced by Lloyd Shapley in the context of cooperative game theory that involves a fair distribution of both gains and costs to several players acting in coalition [40]. The SHAP method tries to ensure that each actor gains as much or more as they would have from acting independently. In this study, we computed the shapley values using the SHAP library [39].

## Discussion

This study introduces a methodology that can be utilized to predict if a ball possession may lead to attacking actions at the last fourth of the soccer field, considering only players' positions and events that occur in the first 5 seconds after the team recovers the ball. The process was developed using different approaches like complex networks (to map the team positions and interactions between players) and image analysis (creating visual rhythm images to represent the time series of the match), so creating a new promising method to classify ball possessions. In this section, we will discuss the top-3 metrics presented in Figs 12 and 13, demonstrating an example and how to interpret each metric.

For the attacking team, the high eccentricity shows a necessity for a player to stay separated, probably improving options for passes and plays. An example of such a situation is presented in Fig 4. But for the team that is defending, this distance between defending players helps the attacking team to arrive in the target zone, due to spaces created. This insight is compatible with other analyses found in the literature of soccer matches [9, 16, 41].

Considering the Local Efficient of the attacking team, this information shows the need for a "central player" responsible for constructing the play, and in a defensive way, the same metric tells us if the defense concentrates on a unique defender, the chances for success to the other team increase.

The high values of Entropy to the attacking team represent that players are free to receive the ball (no opponents near), as illustrated in Fig 6. However, high values considering the defensive team when a player is always available, indicate that he is isolated, creating a deficit in the marking, as shown in Fig 7.

Both teams showed significant SHAP values for the same metrics. This result indicates that both teams ($t_1$ and $t_2$) have importance to the classifier. The Lucey *et al.* work shows something similar, demonstrating that the team without ball possession can influence the play as much as the team with ball [42].

Our results agree with the findings reported by Barreira *et al.* [43], who observed that both the location in the field and the circumstance in which the team gained control of the ball are

essential for scoring goals. That is demonstrated in this work because the network metrics showed up efficient to classifier the BPI.

Tenga *et al.* [44] showed that counterattacks are more effective against tactically non-organized defenses. Counterattacks are also an object of study for Moura *et al.*, in which authors concluded that taking control of the ball in the attacking zone and making few passes to the final shot offers more chance of scoring a goal than taking the ball in the defensive field and going forward performing several passes [45].

As the concept of the framework is established, an open point to future research is to increase the accuracy and generalization capacity. In order to achieve that, one could make use of larger datasets, like all matches of a championship or a complete season. With such kind of data, it would be possible to apply our framework in more varied scenarios.

## Conclusion

Our analysis offers a positive answer to the first research question: "Can we predict if an attacking team will reach the fourth quarter of the field (attacking zone) by considering teams' movements during the first five seconds after a team takes control of the ball?". Notice that the classifier reaches an acceptable balanced accuracy in the test data.

The second research question has also been answered: "Could complex network analysis provide a set of meaningful features to predict if the ball will reach the fourth quarter of the field before the attacking team loses control of the ball?". The SHAP values estimated using the classifiers highlight that eccentricity, local efficiency, and entropy were the most important features, and this finds enough support in the literature.

It is an open question as to whether our framework is effective in other classification problems, such as match result prediction, and classification of passes or includes a windowed approach that includes more than only the first five seconds of a BPI. Another pathway could be the integration of additional (more sophisticated) success criteria, like expected goals. Finally, different ways to create the graph could highlight, considering the passes between players.

## Supporting information

**S1 Data. Analyzed data.** https://doi.org/10.6084/m9.figshare.19222746 (player position and tracking).
(TXT)

**S1 Code. Code utilized.** https://github.com/lstival/soccer_graph_classification (Code repository).
(TXT)

## Acknowledgments

The authors would like to thanks Prof. Dr. Daniel Memmert German Sport University Cologne, São Paulo Research Foundation (FAPESP) and Coordenação de Aperfeiçoamento de Pessoal de Nível Superior—Brasil (CAPES).

## Author Contributions

**Conceptualization:** Leandro Stival, Allan Pinto, Paulo Roberto Pereira Santiago, Ricardo da Silva Torres, Ulisses Dias.

**Data curation:** Leandro Stival, Felipe dos Santos Pinto de Andrade, Paulo Roberto Pereira Santiago, Ricardo da Silva Torres, Ulisses Dias.

**Formal analysis:** Leandro Stival, Allan Pinto, Felipe dos Santos Pinto de Andrade, Ricardo da Silva Torres, Ulisses Dias.

**Funding acquisition:** Paulo Roberto Pereira Santiago, Ricardo da Silva Torres.

**Investigation:** Leandro Stival, Allan Pinto, Felipe dos Santos Pinto de Andrade, Ulisses Dias.

**Methodology:** Leandro Stival, Allan Pinto, Felipe dos Santos Pinto de Andrade, Paulo Roberto Pereira Santiago, Henrik Biermann, Ricardo da Silva Torres, Ulisses Dias.

**Project administration:** Leandro Stival, Allan Pinto, Paulo Roberto Pereira Santiago, Ricardo da Silva Torres, Ulisses Dias.

**Resources:** Leandro Stival, Felipe dos Santos Pinto de Andrade, Ricardo da Silva Torres, Ulisses Dias.

**Software:** Leandro Stival, Allan Pinto, Felipe dos Santos Pinto de Andrade, Ulisses Dias.

**Supervision:** Leandro Stival, Allan Pinto, Paulo Roberto Pereira Santiago, Ricardo da Silva Torres, Ulisses Dias.

**Validation:** Leandro Stival, Felipe dos Santos Pinto de Andrade, Paulo Roberto Pereira Santiago, Henrik Biermann, Ricardo da Silva Torres, Ulisses Dias.

**Visualization:** Leandro Stival, Allan Pinto, Felipe dos Santos Pinto de Andrade, Paulo Roberto Pereira Santiago, Ricardo da Silva Torres, Ulisses Dias.

**Writing – original draft:** Leandro Stival, Allan Pinto, Felipe dos Santos Pinto de Andrade, Paulo Roberto Pereira Santiago, Ricardo da Silva Torres, Ulisses Dias.

**Writing – review & editing:** Leandro Stival, Allan Pinto, Felipe dos Santos Pinto de Andrade, Paulo Roberto Pereira Santiago, Henrik Biermann, Ricardo da Silva Torres, Ulisses Dias.

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
