## [Decision Letter · Decision Letter 0]

24 Jun 2022

PONE-D-22-05975Using machine learning pipeline to predict entry into the attack zone in footballPLOS ONE

Dear Dr. Santiag,

Thank you for submitting your manuscript to PLOS ONE. After careful consideration, we feel that it has merit but does not fully meet PLOS ONE’s publication criteria as it currently stands. Therefore, we invite you to submit a revised version of the manuscript that addresses the points raised during the review process.

We look forward to receiving your revised manuscript.

Kind regards,

Nguyen Quoc Khanh Le

Academic Editor

PLOS ONE

“This work has been supported by the following Brazilian research agencies grants 414 #2019/17729-0, #2019/22262-3, #2020/11946-6, #2019/16253-1, and #2021/00050-4, 415 S˜ao Paulo Research Foundation (FAPESP). This study was financed in part by the 416 Coordena¸c˜ao de Aperfei¸coamento de Pessoal de N´ıvel Superior – Brasil (CAPES) – 417 Finance Code 001.”

Additional Editor Comments:

Please improve the study especially in terms of novelty, algorithm implementation, and case studies.

Reviewers' comments:

Reviewer's Responses to Questions

**Comments to the Author**

1. Is the manuscript technically sound, and do the data support the conclusions?

Reviewer #1: Partly

Reviewer #2: Yes

Reviewer #3: No

2. Has the statistical analysis been performed appropriately and rigorously? 

Reviewer #1: Yes

Reviewer #2: Yes

Reviewer #3: No

3. Have the authors made all data underlying the findings in their manuscript fully available?

Reviewer #1: Yes

Reviewer #2: No

Reviewer #3: Yes

4. Is the manuscript presented in an intelligible fashion and written in standard English?

Reviewer #1: Yes

Reviewer #2: Yes

Reviewer #3: No

5. Review Comments to the Author

Reviewer #1: In this paper authors propose a framework to predict the success of ball possession in football. In particular, for each ball possession they extract a set of features defined relying on complex networks analysis. Then, they exploit Deep Learning techniques to predict a binary outcome for every ball possession. The outcome refers to the event of bringing the ball close to the opponents' goal.

I appreciate the effort of the authors, and I believe this paper would be ready for publication if some issues will be addressed:

- case study: given the nature of the paper, a set of use cases with the relative Shap local explanation are required to provide the reader some results from the application of the proposed methodology. A bit of of analytics would also improve the paper: who are the players mostly contributing with/without the ball to succesful ball possession? What are the network features that defines such players?

- experiment replicability: authors provide both data and code of their work, this is a huge strong point compared with many other papers on the same topic. However, data description is missing, both within the paper and in the data repository. A broader data description would improve this aspect.

Reviewer #2: This paper presented to Use machine learning pipeline to predict entry into the attack zone in football. Overall, the structure of this paper is well organized, and the presentation is relatively clear. The idea is interesting and potential. However, there are still some crucial problems that need to be carefully addressed before a possible publication. More specifically,

1. The motivations or remaining challenges are not so clear or what kinds of issues or difficulties are this task that is facing. Please give more details and discussion about the key problems solved in this paper, which is largely different from existing works.

2. A deep literature review should be given, particularly advanced deep learning models in image processing. Therefore, the reviewer suggests discussing some related works by analyzing the following papers in the revised manuscript.

3. Please clarify the contributions to this field, for example, which are the existing ones and which are your own ones?

4. How about the computational complexity of the proposed method?

5. Some future directions should be pointed out in the conclusion.

Reviewer #3: The Research Paper stands Rejected and is NOT RECOMMENDED for Publication because of the following strong reasons:

1. The Information highlighted and the conceptual methodology is already known and lots of advanced papers are already published.

2. No specific novelty is there.

3. No System model and Algorithm is highlighted in the proposed approach.

4. No Real Time case study based discussion.

6. PLOS authors have the option to publish the peer review history of their article (what does this mean?). If published, this will include your full peer review and any attached files.

Reviewer #1: No

Reviewer #2: No

Reviewer #3: No

---

## [Author Response · Author response to Decision Letter 0]

16 Aug 2022

Thank you for the opportunity to review our manuscript for the PLOS ONE. We appreciate the Reviewer’s constructive comments and through reviews.

The content of the manuscript has been revised to address all the Reviewers’ concerns. After addressing all the comments and revising the manuscript, we are submitting the revised version of our article for your consideration.

All changes made in the manuscript are highlighted. We considered all comments and notes carefully, and we are sure they represent a significant contribution to the quality of our manuscript.

We thank you for this opportunity. Please let us know if there are any further questions or concerns.

Sincerely

Authors

Title:

Using machine learning pipeline to predict entry into the attack zone in football

**Please note that all changes made to the manuscript, legends and figures are highlighted in yellow.

Reviewer 2

1. The motivations or remaining challenges are not so clear, or what kinds of issues or difficulties are this task that is facing. Please give more details and discussion about the key problems solved in this paper, which is largely different from existing works.

R: The main motivation for this work relies on the lack of effective approaches to predict if a ball possession may lead to a chance of scoring. Therefore, in this paper we aimed to create an effective framework that could lead to insights of how the attacking team may approach the penalty box, i.e., may have more chances of scoring. The main novelty of our work relies on the use of a complex network formulation. Other papers analyzed teams using similar metrics to ours, but for different applications and objectives.

The main contribution is to show that the use of complex network features to understand and predict the attack behavior is viable. Another contribution relies on the combination of time series representations with convolutional networks, pre-trained with images, to produce effective predictions. To the best of our knowledge, no study in the literature has addressed this problem using the methodological steps investigated in this paper.

2. A deep literature review should be given, particularly advanced deep learning models in image processing. Therefore, the reviewer suggests discussing some related works by analyzing the following papers in the revised manuscript.

R: The actual machine learning applications cover several areas in football data. Our literature review is basically focused on predictions, and some of these works are described in the Section Related Work in the paper.

Lines 55-133 provide an overview of studies that investigate predictions in soccer context and Lines 294-302 describe how deep learning has been used to predict the tracking of players.

3. Please clarify the contributions to this field, for example, which are the existing ones, and which are your own ones?

R: The contributions presented in our project have proven that it is possible to use complex network features to predict if the team with ball possession will reach the attack zone, i.e., will have chances of scoring. 

4. How about the computational complexity of the proposed method?

R: The complete deep learning network architecture has 6.7 million of trainable parameters, where most of them (approximately 4 millions) belong to EfficientNet (the main backbone used in our formulation). Recall that those parameters (weights) are only fine turned in our trained strategy.

Training the used architecture takes one day on a GPU NVIDIA (GTX 1050 Ti with 4 GB of RAM memory) with CPU (AMD FX-8350-e) and 16 GB of RAM memory. Recall that this machine has a very simple configuration, which demonstrates the advantage of using pre-trained models and simplifying our framework. This training procedure is performed off-line.

The other components of the devised pipeline (e.g., construction of graphs, extraction of metrics from complex network, and the build of visual rhythms) take negligible time when compared to the training time of the network architecture.

5. Some future directions should be pointed out in the conclusion.

R: Thanks for pointing out this issue. In the current version of the manuscript, we extended the discussion upon possible research directions for future work. Possible next steps include the implementation of graph-convolutional networks to improve the results and make the pipeline smaller.

Another possible direction related to incorporating more spatial features related to the attacking and defending team, such as covered area and centroid of the team as Frencken and Lemmink work (FRENCKEN et al., 2011), We believe that the combination of complex network and spatial features may lead to improved prediction results.

Reviewer 3

6. The Information highlighted, and the conceptual methodology, is already known and lots of advanced papers are already published.

R: The paper aims to validate a new approach using concepts already consolidated, like graphs to analyze the players and teams behavior, visual rhythm in time series representation and machine learning to predict the play flow. It is possible that there are works using state-of-the-art algorithms, but our focus was to create and test the possibility of combination of graphs, time series representations and machine learning, in a unique pipeline with few data and heterogeneous teams. To the best of our knowledge, no study in the literature has investigated this combination for similar problems.

7. No specific novelty is there.

R: No algorithm, or metric was the focus of innovation during the development of this project, our attention was kept to create an innovation in the method of how to apply great tools available to predict the movement of the teams. In this paper, we aimed to create an effective framework that could lead to insights of how the attacking team may approach the penalty box, i.e., may have more chances of scoring. The main novelty of our work relies on the using of a complex network formulation perspective. Other papers analyzed teams using similar metrics to ours, but for different applications and objectives. The main contribution is to show that the use of complex network features to understand and predict the attack behavior is viable. Another contribution relies on the combination of time series representations with convolutional networks, pre-trained with images, to produce effective predictions. To the best of our knowledge, no study in the literature has addressed this problem using the methodological steps investigated in this paper.

8. No System model and Algorithm is highlighted in the proposed approach.

R: During the implementation, a neural network was utilized to make the predictions, the focus was to show how that makes it possible to create a pipeline that unify graph, time series, images and deep learning. It describes how a convolutional network is used and the amount of layers between lines 294 and 302.

9. No Real Time case study based discussion.

R: The idea to apply the concepts of this work is to consider the historical data of a time, create a tool to enable the understanding of how a team usually attacks and create a strategy to stop them. Real time application looks a great way to use this pipeline, but it can be a possible continuation of the work. Recall that a real-time application of machine learning algorithms on positional data is uncommon, despite its potential. We believe that is in part due to the enormous computational resources that might be needed.

Reviewer 1

10. Case study: given the nature of the paper, a set of use cases with the relative Shap local explanation are required to provide the reader some results from the application of the proposed methodology. A bit of analytics would also improve the paper: who are the players mostly contributing with/without the ball to successful ball possession? What are the network features that define such players?

R: Thanks for raining such an issue, Shap-based analysis may provide us the values for each channel of image (where each channel represents a metric). It is possible, therefore, to identify and appoint the most important players. However, in the development of the pipeline, this idea was discarded, because the training was realized with data of different teams. We plan to perform individual-based analyses in future work.

11. Experiment replicability: authors provide both data and code of their work, this is a huge strong point compared with many other papers on the same topic. However, data description is missing, both within the paper and in the data repository. A broader data description would improve this aspect.

R: Originally, the description of data was not made available. The data provided is now accompanied with metadata, including for example descriptions of labels and how information is presented in each column: https://figshare.com/articles/dataset/REDSCAT2/19222746

This organization leverages the replicability of our work and allows the validation of our results.

REFERENCES:

FRENCKEN, W.; LEMMINK, K.; DELLEMAN, N.; VISSCHER, C. Oscillations of centroid position and surface area of soccer teams in small-sided games. European Journal of Sport Science, Taylor & Francis, v. 11, n. 4, p. 215–223, 2011.

---

## [Decision Letter · Decision Letter 1]

2 Oct 2022

PONE-D-22-05975R1Using machine learning pipeline to predict entry into the attack zone in footballPLOS ONE

Dear Dr. Santiag,

Thank you for submitting your manuscript to PLOS ONE. After careful consideration, we feel that it has merit but does not fully meet PLOS ONE’s publication criteria as it currently stands. Therefore, we invite you to submit a revised version of the manuscript that addresses the points raised during the review process.

We look forward to receiving your revised manuscript.

Kind regards,

Nguyen Quoc Khanh Le

Academic Editor

PLOS ONE

Reviewers' comments:

Reviewer's Responses to Questions

**Comments to the Author**

1. If the authors have adequately addressed your comments raised in a previous round of review and you feel that this manuscript is now acceptable for publication, you may indicate that here to bypass the “Comments to the Author” section, enter your conflict of interest statement in the “Confidential to Editor” section, and submit your "Accept" recommendation.

Reviewer #1: All comments have been addressed

Reviewer #2: (No Response)

Reviewer #3: All comments have been addressed

2. Is the manuscript technically sound, and do the data support the conclusions?

Reviewer #1: Yes

Reviewer #2: Yes

Reviewer #3: Yes

3. Has the statistical analysis been performed appropriately and rigorously? 

Reviewer #1: Yes

Reviewer #2: Yes

Reviewer #3: Yes

4. Have the authors made all data underlying the findings in their manuscript fully available?

Reviewer #1: Yes

Reviewer #2: Yes

Reviewer #3: Yes

5. Is the manuscript presented in an intelligible fashion and written in standard English?

Reviewer #1: Yes

Reviewer #2: Yes

Reviewer #3: Yes

6. Review Comments to the Author

Reviewer #1: All my comments have been addressed. I appreciate code and data sharing, it's not common among the papers in the field of sports data science.

Reviewer #2: This paper proposed to use machine learning methods to predict entry into the attack zone in football Overall, the structure of this paper is well organized, and the presentation is clear. However, there are still some crucial problems that need to be carefully addressed before a possible publication. More specifically,

1. The motivations or remaining challenges are not so clear or what kinds of issues or difficulties are this task that is facing. Please give more details and discussion about the key problems solved in this paper, which is largely different from existing works.

2. A deep literature reviews should be given, particularly advanced and SOTA deep learning or AI models in data processing and high-level analysis. Therefore, the reviewer suggests discussing some related works by analyzing the following papers in the revised manuscript.

3. What are the main differences between the proposed attention techniques and existing machine learning related methods?

4. The reviewer is wondering how about the computational complexity of the proposed method?

5. It is well-known that the signals or images usually tend to suffer from various degradation, noise effects, or variabilities in the process of imaging. Please give the discussion and analysis. The reviewer is wondering what will happen if the proposed method meets the various variabilities.

6. Some future directions should be pointed out in the conclusion.

Reviewer #3: The Revised Manuscript has incorporated all the revisions and suggestions as mentioned in the last review. And now the paper stands Accepted with no further revisions.

7. PLOS authors have the option to publish the peer review history of their article (what does this mean?). If published, this will include your full peer review and any attached files.

Reviewer #1: **Yes: **Paolo Cintia

Reviewer #2: No

Reviewer #3: No

---

## [Author Response · Author response to Decision Letter 1]

17 Nov 2022

Response to Reviewers

Thank you for the opportunity to review our manuscript for the PLOS ONE. We appreciate the Reviewer’s constructive comments and through reviews.

The content of the manuscript has been revised to address all the Reviewers’ concerns. After addressing all the comments and revising the manuscript, we are submitting the revised version of our article for your consideration.

We considered all comments and notes carefully, and we are sure they represent a significant contribution to the quality of our manuscript.

We thank you for this opportunity. Please let us know if there are any further questions or concerns.

Sincerely

Authors

Title:

Using machine learning pipeline to predict entry into the attack zone in football

**Please note that all changes made to the manuscript, legends and figures are highlighted in yellow.

**This document needs to be compiled in Tex Live Version 2021 of Overleaf. The document will not compile in the new version 2022.

Reviewer 2

1. The motivations or remaining challenges are not so clear or what kinds of issues or difficulties are this task that is facing. Please give more details and discussion about the key problems solved in this paper, which is largely different from existing works.

R: The key problem addressed in this paper refers to mapping features related to the dynamics of players and their actions to relevant events in soccer matches. 

This paper focuses on how to construct an effective representation of teams' dynamics using individual player information. In our formulation, we consider features associated with both attacking and defending teams at the same time.

Our representation of soccer matches stands out for combining techniques never before applied together: complex networks as a consolidated way to observe the behavior of teams; visual rhythms responsible for translating the metrics into a format that makes temporal consistency more latent; and image processing of these features to enable the use of consolidated neural networks to aid in our prediction.

Thus, we can confirm that our main contribution is a novel way to represent, evaluate, and predict scoring opportunities based on features associated with ball possession in soccer matches, using a new approach based on already consolidated concepts to optimize our results.

2. A deep literature review should be given, particularly advanced and SOTA deep learning or AI models in data processing and high-level analysis. Therefore, the reviewer suggests discussing some related works by analyzing the following papers in the revised manuscript.

R: We included some recent studies regarding the mentioned topics in our related works section. In the current version of the manuscript, we describe and discuss how SOTA machine learning models have been applied to sports science problems, particularly soccer-related ones. These descriptions are included between the lines 150-184.

3. What are the main differences between the proposed attention techniques and existing machine learning related methods?

R: In our process, we used a pre-trained network called EfficientNet B0, where we fine-tuned its predictions to suit our needs.

Thus, our major contribution is not aimed at a new model or architecture, but rather the creation of a pipeline where multichannel images were processed by this network originally pre-trained for three-channel images. As achieved results suggest, the use of the proposed pipeline leads to very effective results in the considered prediction problem. Recall that the proposed pipeline is flexible and can easily accommodate the use of other architectures. The use of more effective pre-trained architectures would lead to even better results.

4. The reviewer is wondering how about the computational complexity of the proposed method?

R: It took 1.5 Gb of vram to train the 44 million parameters, including fine-tuning the EfficientNet B0. Also, to obtain the graph metrics, we used the methods provided by the python library called networkx, and it required approximately 4.2 seconds to generate the metrics for each BPI. More details of the configuration used for training (such as time and hardware) are presented in the manuscript on the topic Computational Resources described in lines 362 -369.

5. It is well-known that the signals or images usually tend to suffer from various degradation, noise effects, or variabilities in the process of imaging. Please give the discussion and analysis. The reviewer is wondering what will happen if the proposed method meets the various variabilities.

R: In our approach, we create our images from the Visual Rhythms obtained from the translation of the values computed from each metric into a pixel for the corresponding image. We understand that once there are several channels for each pixel (the frames in a sequence) the noise to the pixel value is not significant enough to affect the quality of the image.

6. Some future directions should be pointed out in the conclusion.

R: As open lines left by our research, we can mention the investigation of features related to the team formation, such as, for example, the centroid or area of a team. Future work would then be concerned with the assessment of the use of this information for predicting relevant soccer events.

Another research direction refers to directly extracting features from the graphs instead of computing image representations, such as visual rhythms. One suitable alternative would be to employ methods based on convolutional graph networks.

Second awnsers

1. Please note that funding information should not appear in the Acknowledgments section/Funding section or Any other areas of your Manuscript. We will only publish funding information present in the Funding Statement section of the online submission form. Please remove any funding-related text from the manuscript.

R: We have adjusted the Acknowledgements and removed all the funding information there.

2. Please provide additional details regarding participant consent. In the Methods section, please ensure that you have specified (1) whether consent was informed and (2) what type you obtained (for instance, written or verbal). If your study included minors, state whether you obtained consent from parents or guardians. If the need for consent was waived by the ethics committee, please include this information.

R: Our project was approved by the Research Ethics Committee of the State University of Campinas, protocol CAAE 56582616.8.0000.5404. And since all those present are of legal age, the need for consent was waived by the ethics committee, due to the video recording being done with proper cameras installed in the stadium in professional games

---

## [Decision Letter · Decision Letter 2]

1 Dec 2022

Using machine learning pipeline to predict entry into the attack zone in football

PONE-D-22-05975R2

Dear Dr. Santiag,

We’re pleased to inform you that your manuscript has been judged scientifically suitable for publication and will be formally accepted for publication once it meets all outstanding technical requirements.

Kind regards,

Nguyen Quoc Khanh Le

Academic Editor

PLOS ONE

Additional Editor Comments (optional):

Reviewers' comments:

Reviewer's Responses to Questions

**Comments to the Author**

1. If the authors have adequately addressed your comments raised in a previous round of review and you feel that this manuscript is now acceptable for publication, you may indicate that here to bypass the “Comments to the Author” section, enter your conflict of interest statement in the “Confidential to Editor” section, and submit your "Accept" recommendation.

Reviewer #2: All comments have been addressed

2. Is the manuscript technically sound, and do the data support the conclusions?

Reviewer #2: Yes

3. Has the statistical analysis been performed appropriately and rigorously? 

Reviewer #2: Yes

4. Have the authors made all data underlying the findings in their manuscript fully available?

Reviewer #2: Yes

5. Is the manuscript presented in an intelligible fashion and written in standard English?

Reviewer #2: Yes

6. Review Comments to the Author

Reviewer #2: The authors have well addressed the reviewer's comments. No more comments. It is ready to accept the current manuscirpt.

7. PLOS authors have the option to publish the peer review history of their article (what does this mean?). If published, this will include your full peer review and any attached files.

Reviewer #2: No

---

## [Editor Report · Acceptance letter]

27 Dec 2022

PONE-D-22-05975R2 

Using machine learning pipeline to predict entry into the attack zone in football 

Dear Dr. Dias:

I'm pleased to inform you that your manuscript has been deemed suitable for publication in PLOS ONE. Congratulations! Your manuscript is now with our production department. 

Kind regards, 

on behalf of

Dr. Nguyen Quoc Khanh Le 

Academic Editor

PLOS ONE